# A Novel Ionospheric Sounding Network Based on Complete Complementary Code and Its Application

**DOI:** 10.3390/s19040779

**Published:** 2019-02-14

**Authors:** Tongxin Liu, Guobin Yang, Yaogai Hu, Chunhua Jiang, Ting Lan, Zhengyu Zhao, Binbin Ni

**Affiliations:** School of Electronic Information, Wuhan University, Wuhan 430072, China; tongxin_liu@whu.edu.cn (T.L.); chuajiang@whu.edu.cn (C.J.); tinglan@whu.edu.cn (T.L.); zhaozy@whu.edu.cn (Z.Z.); bbni@whu.edu.cn (B.N.)

**Keywords:** ionospheric sounding network, complete complementary code, ionogram inversion, data assimilation

## Abstract

In this paper, complete complementary code (CCC) sequences are applied to a High Frequency (HF) ionospheric sounding network. Ionosondes distributed at multiple locations use the mutually orthogonal CCC sequences to conduct vertical soundings synchronously. At the same time, thanks to the omnidirectional antennas, every station can receive the oblique echoes transmitted from the others. Due to the orthogonality between the code sequences, both vertical and oblique ionograms can be simultaneously obtained and completely separated. Through this method, the sounding efficiency can be enhanced, and the inversion difficulty can be reduced. Further, by using the data assimilation method, vertical and oblique sounding results can be combined to obtain a wide range of regional ionospheric characteristics. To verify the performance of this kind of sounding network, validation experiments are implemented to demonstrate that vertical and oblique ionograms can be obtained independently at the same time and inverted separately and that the maps of foF2 parameters obtained by using the data assimilation method provide more details than single vertical or oblique sounding.

## 1. Introduction

With the maturity of traditional sounding technology and the recent advances of digital, low-cost radio equipment, the ionospheric sounding mode is also developing towards multi-station networking [1]. Compared with the monostatic mode, many ionospheric characteristics at different locations can be obtained synchronously with multistatic network that can be very helpful to do the regional ionospheric research. In general, ionospheric sounding network can be divided into vertical and oblique network with its own advantages. 

As one of the earliest, mostly used ground-based sounding methods, vertical ionospheric sounding plays an important role in ionospheric research [2,3]. For vertical sounding network, a commonly used method is to deploy an appropriate number of ionosondes to cover the interested area to study regional ionospheric characteristics. However, there are some limited conditions. Firstly, it is difficult to deploy ionosondes adequately to ensure the accuracy. Secondly, some places, such as mountain areas or lakes, might not be appropriate for installing ionosondes. In many cases, ionospheric oblique sounding is an effective substitute solution [4,5]. As ionosondes can receive echoes from other stations and obtain the midpoints’ ionospheric characteristics of the propagation paths, there is no necessity to install ionosondes at unsuitable places. To date, there are two main methods for ionospheric oblique sounding network [6]. One is to use chirp waveform, such as the ionospheric oblique-incidence sounding network in Russia [7], single transmitter and multiple receiver systems like Inskip-Rome, Inskip-Chania oblique-incidence sounding network [5], and the system based on the combination of pseudorandom phase coding and chirp modulation waveform [6]. However, there is a limitation of using chirp waveform. As a pure oblique sounding method, it would miss the local ionospheric information of transmitters. Another oblique sounding method is based on the vertical sounding systems by using omnidirectional antennas, such as vertical-incidence sounding network developed by Massachusetts Lowell University [8] and the DPS4D system set up in Europe. Due to the omnidirectional nature of the antennas, these systems can receive vertical and oblique echoes simultaneously. The observation and research results on travelling ionospheric disturbances (TIDs) based on the oblique sounding network of DPS4D system were reported by Tobias Verhulst et al. in 2017. However, when the distance between two stations is not suitable, the low-frequency part of the oblique echoes can easily overlap with the vertical echoes or its multi-hop, which will be well illustrated below. Additionally, since the echoes of different transmitters are usually coded in the same way, it is difficult to distinguish the signals of different transmitting sources. Although DPS4D can overcome this problem by distinguishing vertical and oblique signals by detecting the directions of the received echoes, it requires four receiving antennas to form an array [9].

In order to overcome the limitations of the above sounding network, an innovative networking method based on complete complementary code (CCC) is proposed in our study. We employ the mutually orthogonal CCC sequences on the ionosondes arranged at multiple places to carry out synchronizing vertical soundings. By employing an omnidirectional antenna at each receiver, every ionosonde can also receive the oblique signals transmitted by the others. With no need for arrays, relaying on the orthogonality between the CCC sequences, the signals of different transmitting sources can be completely separated. Therefore, ionograms of vertical and oblique soundings can be obtained without aliasing. In this way, the difficulty of ionogram inversion can be greatly reduced. For data assimilation, this method obtains more ionospheric information with fewer stations, thereby being very helpful for improving the accuracy of the results and simplifying the system structure. Accordingly, the validation experiments were carried out in China between June and August 2018. The sounding stations were built at Wuhan (30°32′24″ N, 114°21′12″ E) in Hubei province, Leshan (29°33′36″ N, 103°44′60″ E) in Sichuan province, and Ningqiang (32°52′12″ N, 106°14′24″ E) in Shanxi province to form a sounding network. By employing three different sequences that belong to one CCC set, the experiments achieved the synchronous ionospheric soundings in a multi-station and multi-mode manner effectively. With no extra hardware costs, more accurate maps of foF2 parameters were obtained. 

## 2. Principle

Complete complementary code is a kind of code set composed of several mutually orthogonal complementary code sequences, which is widely used in code division multiple access (CDMA) engineering structure [10,11,12,13,14]. For a set of CCC consisting of M sequences with N orders (where M is the number of sequences in a sequence set, N is the number of subsequences of every sequence), it can be expressed as $C=\{c_{m,n},1\leq m\leq M,1\leq n\leq N\}$, where every order has L bits: $c_{m,n}=(c_{m,n}^{1},c_{m,n}^{2},\cdots ,c_{m,n}^{L})$. For the sounding system that adopts the CCC sequence, its transmitting waveform can be expressed as Equation (1).
$$u_{c(m,n)}(t)=\{\begin{array}{c} \frac{1}{\sqrt{L}}\sum_{k=0}^{L-1}c_{m,n}^{k}u_{1}(t-kt_{p}),0<t<Lt_{p}\\ 0,else \end{array},$$
(1)$$u_{1}(t)=\{\begin{array}{c} \frac{1}{\sqrt{t_{p}}},0<t<t_{p}\\ 0,else \end{array},$$
where L is the sequence length, $c_{m,n}^{k}$ represents the kth bit of the nth order in mth sequence of the CCC set, and $t_{p}$ is the duration of each bit.

If $u_{2}(t)$ is recorded as Equation (2):
(2)$$u_{2}(t)=\frac{1}{\sqrt{L}}\sum_{k=0}^{L-1}c_{m,n}^{k}\delta (t-kt_{p}),$$
$u_{c(m,n)}(t)$ can be evolved into Equation (3):
(3)$$u_{c(m,n)}(t)=\frac{1}{\sqrt{L}}\sum_{k=0}^{L-1}c_{m,n}^{k}u_{1}(t-kt_{p})=u_{1}(t)⊗u_{2}(t)$$

For each order of one code sequence in a set, its ambiguity function is:
(4)$$\begin{array}{c} \chi_{u}(\tau ,\xi )=\int_{-\infty }^{+\infty }u_{c}(t)u_{c}(t+\tau )e^{j2\pi \xi t}dt=\int_{-\infty }^{+\infty }u_{c}(t)e^{j2\pi \xi t}u_{c}(\tau -(-t))dt\\ =u_{c}(\tau )e^{j2\pi \xi t}⊗u_{c}(-\tau )\\ =\left[u_{1}(\tau )e^{j2\pi \xi \tau }⊗u_{1}(-\tau )\right]⊗\left[u_{2}(\tau )e^{j2\pi \xi \tau }⊗u_{2}(-\tau )\right]\\ =\chi_{1}(\tau ,\xi )⊗\chi_{2}(\tau ,\xi ), \end{array}$$
where $\chi_{1}(\tau ,\xi )$ and $\chi_{2}(qt_{p},\xi )$ can be expressed as:
(5)$$\chi_{1}(\tau ,\xi )=\{\begin{array}{c} e^{j\pi \xi (t_{p}-\tau )}\frac{\sin(\pi \xi (t_{p}-\left|\tau \right|))}{\pi \xi (t_{p}-\left|\tau \right|)}(\frac{t_{p}-\left|\tau \right|}{t_{p}}),\left|\tau \right|<t_{p}\\ 0,else \end{array},$$
(6)$$\chi_{2}(qt_{p},\xi )=\{\begin{array}{c} \frac{1}{L}\sum_{k=0}^{L-1-q}c_{k}c_{k+q}e^{j2\pi \xi kt_{p}},0\leq q\leq (L-1)\\ \frac{1}{L}\sum_{k=-q}^{L-1-q}c_{k}c_{k+q}e^{j2\pi \xi kt_{p}},-(L-1)\leq q\leq 0 \end{array}$$

The ambiguity function of each code sequence is the sum of its orders’, and can be formulated as Equation (7):
(7)$$\chi (\tau ,\xi )=\sum_{m=1}^{N}\chi_{m}(\tau ,\xi )$$

For the correlation function, the CCC sequences should satisfy Equation (8), where q is the bit shift. As the cross-correlation result of any pair sequences is zero at all shifts, there is an ideal cross-correlation characteristic between the sequences in CCC set [15,16]. It guarantees the orthogonality between the sequences.
(8)$$\sum_{n=1}^{L}R(c_{m,n},c_{p,n},q)=\{\begin{array}{c} LN,m=p,q=0\\ 0,else \end{array}$$

In detail, Table 1 shows a bi-phase encoding scheme encoded in the CCC manner with the parameters of length L=16, order N=4 as an example. In the experiments described below, this scheme is also used with $t_{p}=25.6\;\mu s$. The simulation results of one sequence’s normalized ambiguity function is plotted in Figure 1. In Table 1, + and −, respectively, represent the phase 0 and π. The four sequences are indexed by letters A, B, C, D, and the orders in a sequence are indexed by numbers 1, 2, 3, 4. In Figure 1, (a) is a three-dimensional ambiguity function graph, (b) is the contour map corresponding to (a), and (c) and (d) are zero Doppler and zero shift sections. Obviously, the CCC has a “pushpin type” ambiguity function, similar to the Pseudo Random Code (PRN) sequences. This means that the CCC can provide a high resolution in distance and speed and a good sounding accuracy. It is very suitable for soft target sounding applications such as the ionospheric sounding described in this paper.

The simulation of the auto-correlation functions and cross-correlation functions of the example scheme is shown in Figure 2 in which (a) is the auto-correlation result of each sequence, and (b) is the cross-correlation result for every pair in one set. It is seen that the CCC’s auto-correlation functions only have the main lobe with the amplitude four times of the code length. Meanwhile, it gives a gain of 36.12 dB. The cross-correlation functions of every pair of the sequences are zero in whole. With the same hardware costs, the only need is to decode the echoes with the corresponding sequences to distinguish the signal sources. It is a good choice for the ionospheric sounding network with multiple stations to achieve the echoes discrimination. 

The specific method for generating CCC sequences and the signal waveform analysis is shown in the literature [17] published previously. In this paper, we mainly introduce the application of this code system in ionospheric sounding network and illustrate the results of validation experiments.

## 3. Networking Method and Experiments

In view of the advantages of CCC, we propose an ionospheric sounding network method based on this code system. Firstly, we upgrade the original ionospheric vertical sounding systems developed by Wuhan University [18] to adapt CCC. Then, mutually orthogonal sequences of one set are assigned to the sounding stations at different places. By using omnidirectional antennas for both transmitting and receiving, when the ionosondes operate synchronously, each station can receive both vertical and oblique echoes at the same time. According to the orthogonality between the code sequences, signals transmitted by different transmitters can also be distinguished by decoding correspondingly. Therefore, the ionograms of the different stations’ vertical soundings and the oblique soundings with different propagation paths are obtained without aliasing, that is, the vertical and oblique soundings can work simultaneously and independently. With no extra hardware costs, it conveniently improves the sounding efficiency. The following experiments are taken as examples to further illustrate this sounding method.

To confirm the practicality and application effect of this method, we conducted verification experiments between June and August 2018. In these experiments, we set three ionosondes with the same construction at Wuhan (30°32′24″ N, 114°21′12″ E), Leshan (29°33′36″ N, 103°44′60″ E) and Ningqiang (32°52′12″ N, 106°14′24″ E). The distribution of the stations is shown in Figure 3. The distances between the stations were, respectively, 1038 km, 827 km, and 428 km. The ionosondes were set up to work in the frequency sweeping mode. All the antennas for transmitting and receiving have the omnidirectional patterns. The specific sounding parameters are shown in Table 2. 

Two experiments were carried out. In the first case, the same sequence of general complementary code was employed to conduct a synchronous sounding at Ningqiang and Wuhan stations as a control experiment. In the second case, we set up three stations to execute synchronous sounding network by employing the CCC sequences. The sequences A, B and C exemplified in the Section 2 were assigned to Wuhan, Leshan and Ningqiang stations respectively. The experimental results will be detailed in Section 4.

It is noted that to ensure the time-frequency synchronization of the networking, the ionospheric sounding systems developed by Wuhan University are embedded with GPS signal receiving modules. As the frequency source of the system can be calibrated by the PPS signal of GPS, the working sequences of each station can be aligned.

Through this method, the expected results of the experiments are that after one run of frequency sweeping, this sounding network can finish the vertical soundings of the three stations and six oblique soundings between them. The ionospheric information of three vertical sounding stations and three midpoints of the oblique sounding propagation paths can be obtained synchronously (for the same path, the reflection points of the two-way propagation are very close). That is, the ionospheric information of more locations can be obtained by only upgrading the code system on the existing ionosondes without extra costs. This advantage can be more significant with the increase of the number of sounding stations and the oblique sounding paths.

In addition, due to the orthogonality between the CCC sequences, since the vertical and oblique ionograms can be completely separated, the difficulty of the ionogram inversion can be greatly reduced. When the vertical and oblique ionograms are inverted respectively, compared with the pure vertical or oblique sounding networking, it is equivalent to insert intermediate data of ionospheric parameters between any two stations. Therefore, we can obtain ionospheric characteristics of six locations instead of the three obtained by pure vertical or oblique sounding network. Consequently, the accuracy of regional ionospheric data assimilation can be potentially improved.

## 4. Results and Discussion

In the first case, when the same code sequence is used, as the vertical and oblique soundings are carried out simultaneously, stations on both sides of one propagation path can receive their own vertical echoes and the oblique echoes at the same time. By taking the station in Ningqiang as an example, Figure 4 displays the ionograms recorded at 14:37 Beijing time (BJT) on 23 June 2018. As previously anticipated, the ionospheric information of multiple locations was obtained at the same time. The local vertical echoes of Ningqiang are distributed at 4–8 MHz. According to the delay time, we can calculate that the virtual height of Es layer in Ningqiang is about 126 km and that of F2 is about 326 km. The oblique signals transmitted by Wuhan and received by Ningqiang are distributed in the range of 7–11 MHz. The propagation distance of the signals reflected by the F2 layer of the propagation midpoints is about 1083 km and that of the Es layer is 891 km. According to the ground distance between Wuhan and Ningqiang, the virtual height of F2 layer and Es layer of the propagation midpoint can be calculated as 349 km and 165 km respectively.

However, because the electromagnetic environment is both frequency and location dependent, the signal strength comparison between the oblique and vertical echoes are not constant. As indicated by point A in Figure 4, the SNR is 41.09 dB as the strongest vertical echo. But the SNR indicated by point B is only 15.98 dB, obviously smaller than those of F2 oblique echoes, 32.42 dB and 28.45 dB, indicated by points C and D (the oblique echo of Es). This may cause large error during the automatic inversion of the vertical echoes.

Figure 4 also indicates that the vertical and oblique echoes are aligned at the low-frequency part. Especially in the range of 6–8 MHz, it is even difficult to distinguish the oblique echoes from the two-hop echoes of the vertical sounding. It may seriously interfere with the automatic inversion of the ionospheric oblique echoes. This phenomenon has a more serious impact when the two stations are close to each other. Besides, if the number of workstations is bigger, the same code sequences used for sounding synchronously can further complicate the ionograms. 

In the second case, the interference is not a problem as demonstrated in Figure 5 which shows the vertical ionograms recorded at 22:06 BJT on 31 August 2018 in which (a) is the vertical ionogram of Leshan station, (b) is the result of Ningqiang, and (c) is for Wuhan. When three mutually orthogonal CCC sequences were used for synchronous soundings at all stations, the vertical ionograms could be obtained separately at the same time. It is obvious that due to the mutually orthogonal characteristic of the CCC sequences of the same set, after the decoding by the local transmitting sequences, pure ionograms can be obtained with no interference of the oblique sounding echoes.

Therefore, the automatic inversions of the vertical ionograms become much easier. Figure 6 shows the automatic inversed current electron density profile of each station based on quasi-parabolic segment (QPS) model. Where (a) is the result of electron density profile according to the vertical ionogram of Leshan, (b) and (c) are the results of Ningqiang and Wuhan respectively.

Moreover, the oblique iongrams can also be obtained due to the omnidirectional antennas. Figure 7 shows the oblique ionograms of each station by the decoding with the other stations’ CCC sequences. In the plots, (a) and (b) are the oblique ionograms based on the received signals of Leshan and decoded by the transmission sequences of Ningqiang and Wuhan. (c) and (d) are the decoded results of Ningqiang with the Leshan and Wuhan’s transmitting sequences respectively. (e) and (f) are the oblique ionograms of Wuhan station according to the code sequences of Leshan and Ningqiang. As seen in Figure 7, in the case of multi-station synchronous sounding and using the mutually orthogonal sequences, a sounding operation can obtain multiple oblique ionograms with no interference. 

Thus, vertical and oblique ionograms and oblique ionograms of different propagation paths do not interfere with each other. Compared with the single vertical or oblique soundings, the ionospheric information and the covering range is greatly increased. The oblique echoes of (a) and (c), (b) and (e), (d) and (f) have the similar pattern of the signal delay time and trace. It also confirms that the reflection points of the two-way propagation of the oblique soundings are very close. 

To study the characteristics of ionospheric sounding propagation paths, this method also provides a good experimental scheme. As the signals from different sources can be well separated by using CCC sequences, the only need for studying multiple propagation paths is to arrange the ionosondes at the terminals of the paths to work in a synchronous mode. There is no need to consider the possible interference.

With a view that the oblique echoes of the E layer and the Es layer are not easy to distinguish in the ionograms, the correct profiles of E or Es layer are difficult to obtain. Therefore, in this paper, we set the F layer parts as the main objects of the oblique inversions. Oblique inversions are also conducted only for F layer. As the reflection points of the two-way propagation of the oblique soundings are very close and based on the assumption that the parameters of the ionosphere are similar in a small region, we select three oblique ionograms from Figure 7 to represent the oblique paths. Figure 8 presents the electron density profiles of the midpoints of the oblique sounding propagation paths by using hybrid genetic algorithm (HGA) based on QP model and the monolayer pattern, in which (a) is the profile inversed from the oblique ionogram of the Leshan-Ningqiang path, and (b) and (c) are the profiles of Leshan–Wuhan and Wuhan–Ningqiang paths. 

HGA is an improved genetic algorithm (GA) based on simulated annealing algorithm (SA). It was firstly applied to the inversion of oblique ionogram by Song Huan et al. [19]. Because of the combination of the advantages of the two algorithms, HGA has high accuracy and efficiency [20]. When it is used in oblique inversion, the results of Song Huan’s experiments show that HGA performs better in accuracy and stability than GA and SA. The specific inversion process and the comparison between the parameters (especially the comparison of the foF2 parameters) obtained by using this algorithm on the oblique inversion and the vertical sounding results at the reflection point can refer to [19].

Since the CCC sequences can acquire more ionospheric information without adding additional ionosondes, compared to the general sounding network methods, instantaneous ionospheric regional characteristics distribution in a large range (like foF2 parameters as an example) can be obtained more accurately through the data assimilation which is essential to studying the spatial characteristics of the ionosphere. 

Figure 9 shows the regional maps of foF2 parameters extracted from the above inversion results through Kriging (KG) algorithm which has been widely used to reconstruct maps of ionospheric parameters [21,22,23]. When the spare date set is abundant, KG has good robustness [24]. (Although different models are used in vertical and oblique inversion, which may cause some inconsistencies between the electron density profiles, but as described in the article [19], the inversion results of foF2 parameters are quite reliable. Referring to the vertical sounding results of the reflection point, 84.62% of the oblique inversion results are within the error of 0.4 MHz). In the plots, (a) is the map assimilated according to the data of the three vertical soundings, (b) is the map based on the data of the three oblique soundings, and (c) is the map involving both the vertical soundings and the oblique soundings. The black triangles are marked as the positions of the sounding stations, while the black circles are marked as the positions of the midpoints of the oblique sounding propagation paths. Figure 2a,b indicate that the foF2 parameters of this region are characteristic of the high west and low east and the high south and low north. However, because the assimilation source is the data of vertical soundings, (a) is more inclined to reflect the foF2 distribution characteristics of the boundary conditions, while (b) is more focused on the middle part. (c) is the synthesis of (a) and (b), and therefore it is more accurate and shows more details. Both of the boundary characteristics and the details of the middle part are included. Compared to the method using vertical or oblique sounding data only, this method of the sounding network can enrich the diversity of the data of foF2 parameters without extra stations. The inverted distribution map of foF2 conforms to the actual situation in a better way.

## 5. Conclusions

In this paper, an ionospheric sounding network method based on complete complementary code is proposed. Based on the orthogonality of the code system, the vertical and oblique sounding network becomes simpler and more efficient. Both vertical and oblique ionograms can be obtained simultaneously and separately. By using the data assimilation method, more sophisticated ionospheric maps can be obtained to provide more detailed and precise spatial characteristics of the coverage area. Since no extra hardware costs is required, this sounding network has great flexibility. In terms of proper networking mode, this method has the important potential for applications into the research of locating ionospheric irregularities, monitoring travelling ionospheric disturbances (TIDs), and even enhancing the global navigation satellite systems (GNSS) by providing regional ionospheric reference information.

However, there remain limitations for the ionospheric sounding network method proposed in this paper. For complete complementary code, the number order is a barrier to the richness of available sequences. As described in this paper, only four mutually orthogonal CCC sequences are included in a 4-order set. This may limit the scale of the sounding network. By increasing the order, more CCC sequences can be available but at the expense of extending the sounding period. Usually, we can treat this issue by reducing the number of coherent accumulations. The SNR loss can be supplemented by the additional gain of increasing code gain. Thus, the adoption of higher code order or other signal patterns with orthogonal characteristics is a subject left for our future investigation in order to organize larger-scale sounding networks.

## Figures and Tables

**Figure 1 sensors-19-00779-f001:**
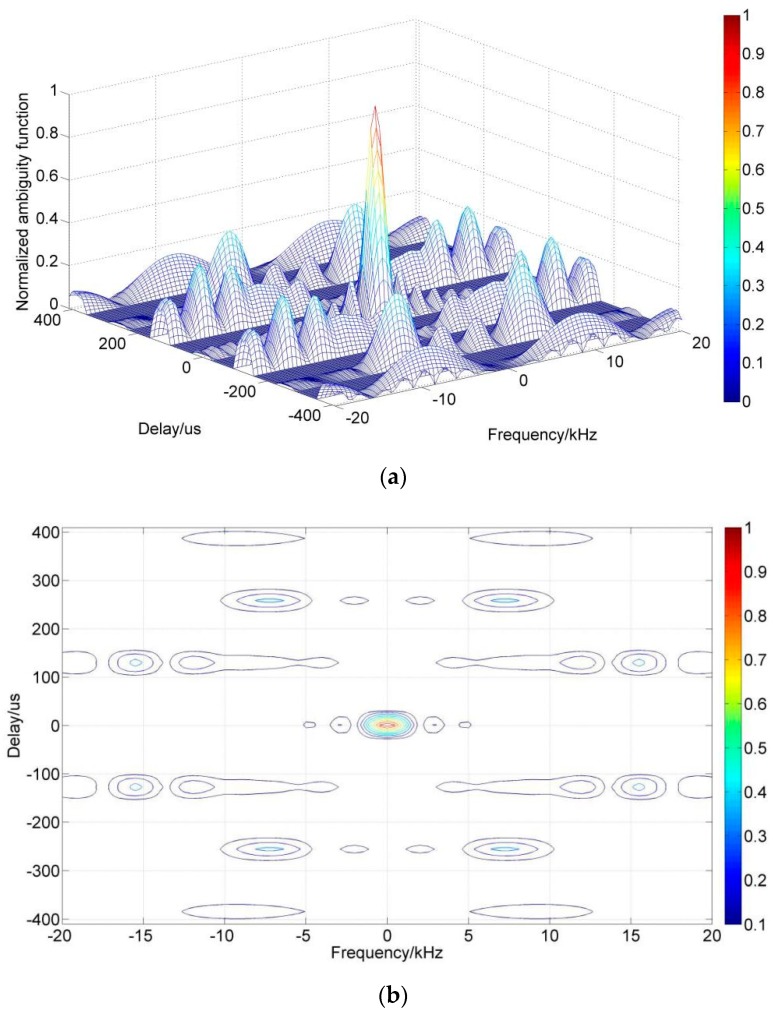
Normalized Ambiguity Function Analysis of Complete Complementary Code: (**a**) Normalized ambiguity function; (**b**) Contour map corresponding to (**a**); (**c**) Normalized ambiguity cut for Frequency = 0 kHz; (**d**) Normalized ambiguity cut for Delay = 0 μs.

**Figure 2 sensors-19-00779-f002:**
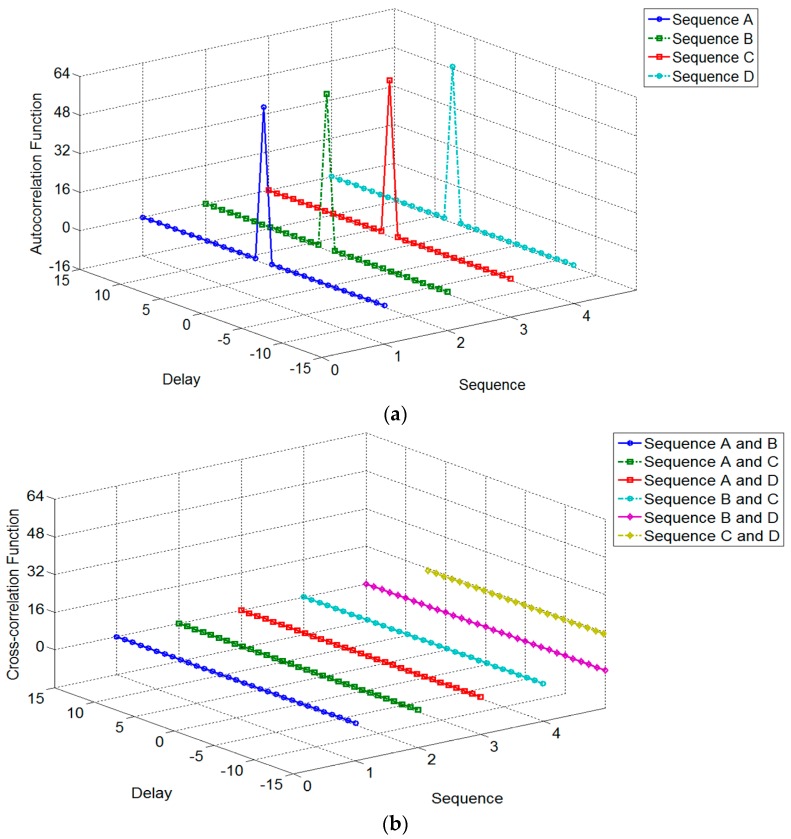
Simulation of the correlation functions: (**a**) Auto-correlation result of each sequence; (**b**) Cross correlation result for every pair in one set.

**Figure 3 sensors-19-00779-f003:**
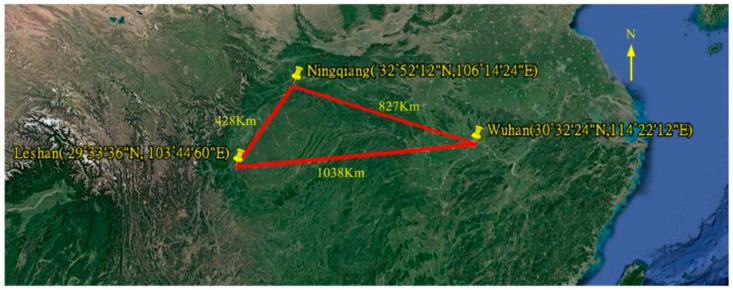
Workstation distribution map.

**Figure 4 sensors-19-00779-f004:**
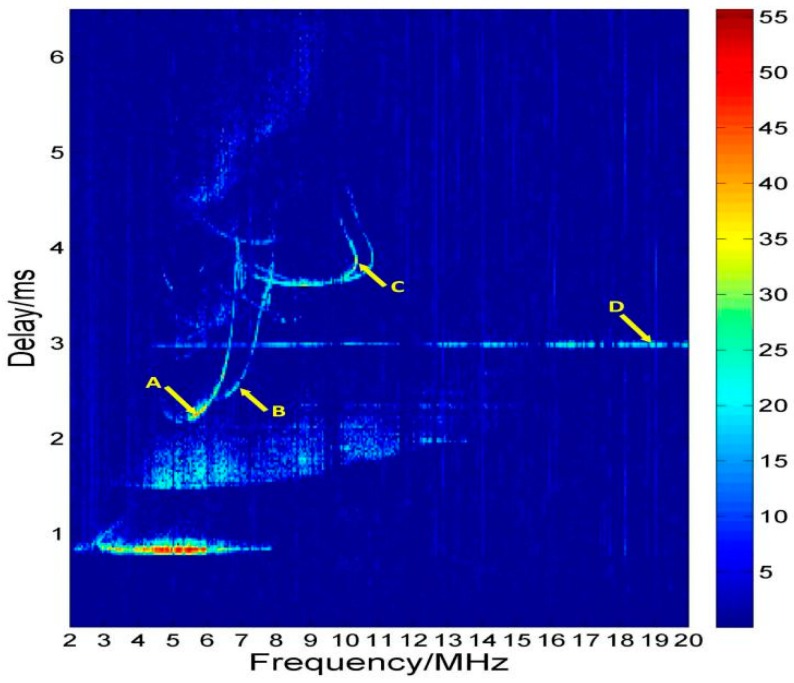
Ionogram of the same complementary code sequence.

**Figure 5 sensors-19-00779-f005:**
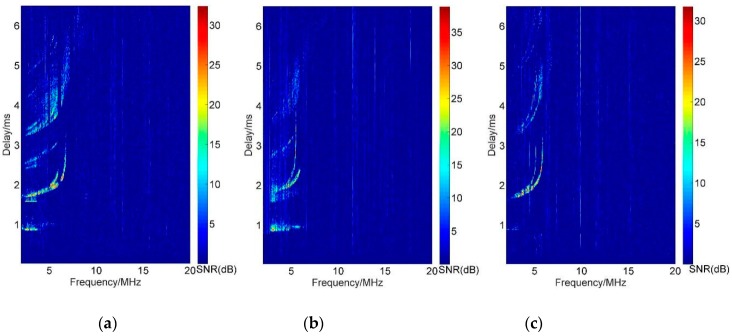
Ionogram of complete complement code sequences: (**a**) Vertical ionogram of Leshan; (**b**) Vertical ionogram of Ningqiang; (**c**) Vertical ionogram of Wuhan.

**Figure 6 sensors-19-00779-f006:**
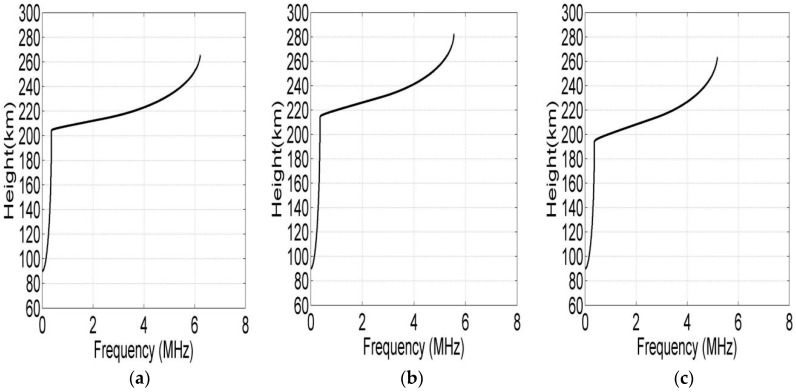
Inversion results of vertical ionograms: (**a**) Electron density profile inversion of Leshan; (**b**) Electron density profile inversion of Ningqiang; (**c**) Electron density profile inversion of Wuhan.

**Figure 7 sensors-19-00779-f007:**
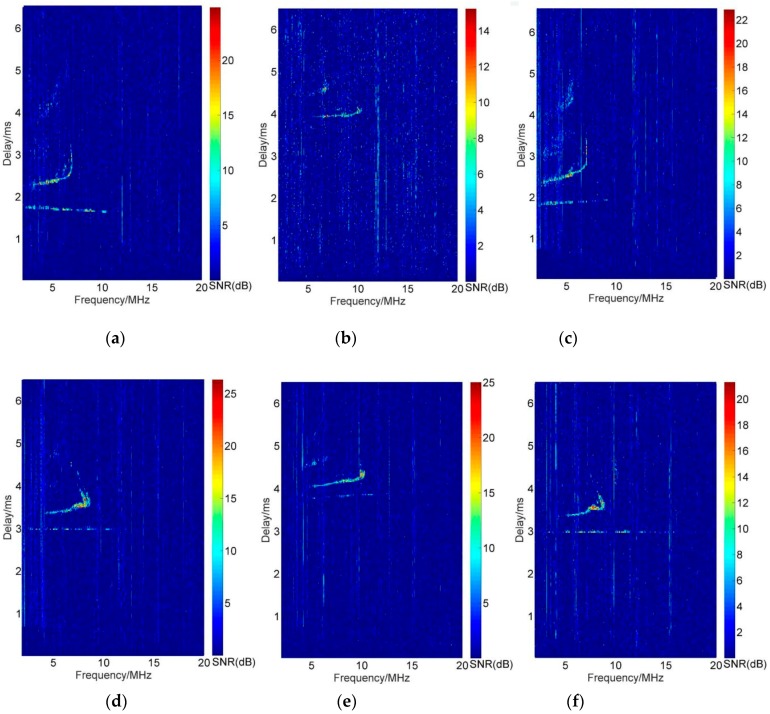
Oblique ionograms of manually orthogonal complete complement sequences: (**a**,**b**) are oblique ionograms of the received signals of Leshan decoded by the sequences of Ningqiang and Wuhan [16]. (**c**,**d**) are the results of the received signals of Ningqiang decoded by the sequences of Leshan and Wuhan respectively. (**e**,**f**) are the ionograms of Wuhan decoded by the sequences of Leshan and Ningqiang.

**Figure 8 sensors-19-00779-f008:**
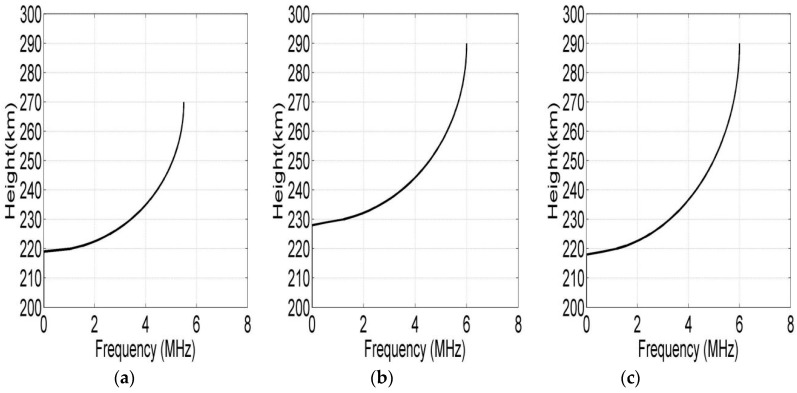
Inversion results of oblique ionograms: (**a**) is the electron density profile inversed from the oblique ionogram of the Leshan–Ningqiang channel. (**b**,**c**) are the electron density profiles of Leshan–Wuhan and Wuhan–Ningqiang.

**Figure 9 sensors-19-00779-f009:**
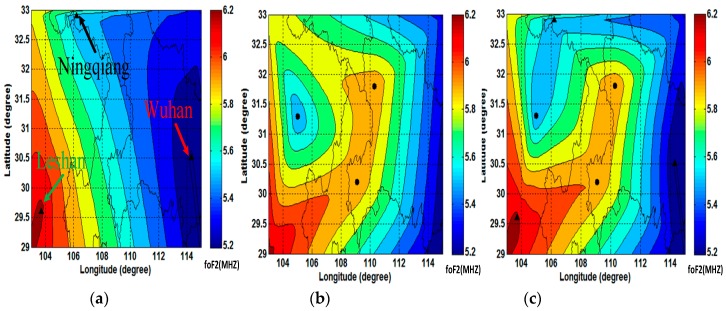
foF2 maps: (**a**) is the map reconstructed according to the data of only three vertical-incidence soundings, (**b**) is based on the three oblique-incidence soundings’ data, and (**c**) is the map involving the vertical-incidence soundings and the oblique-incidence soundings.

**Table 1 sensors-19-00779-t001:** Complete Complementary Code scheme.

**A1**	+ + + + + − + − + + − − + − − +
**A2**	+ − + − + + + + + − − + + + − −
**A3**	+ + − − + − − + + + + + + − + −
**A4**	+ − − + + + − − + − + − + + + +
**B1**	+ + + + − − − − + − − + − + + −
**B2**	+ − + − − − − − + − − + − − + +
**B3**	+ + − − − + + − + + + + − + − +
**B4**	+ − − + − − + + + − + − − − − −
**C1**	+ + + + + − + − − − + + − + + −
**C2**	+ − + − + + + + − + + − − − + +
**C3**	+ + − − + − − + − − − − − + − +
**C4**	+ − − + + + − − − + − + − − − −
**D1**	+ + + + − + − + − − + + − + + −
**D2**	+ − + − − − − − − + + − + + − −
**D3**	+ + − − − + + − − − − − + − + −
**D4**	+ − − + − − + + − + − + + + + +

**Table 2 sensors-19-00779-t002:** Parameters of the experimental system.

System Parameter	Value
Radiated power	200 (W)
Frequency range	220 (MHz)
Frequency step	50 (kHz)
Code system	Complete complementary code
Pulse width	25.6 (μs/bit)
Duty cycle	5%
Coherent accumulation times	32

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
