# Peer review of "A Novel Ionospheric Sounding Network Based on Complete Complementary Code and Its Application"

_sensors, 2019, doi:10.3390/s19040779_

Round 1

Reviewer 1 Report

see the attached file containing review, please

Author Response

Dear Editors and Reviewers:

Thank you for your letter and for the reviewers’ comments concerning our manuscript entitled “A Novel Ionospheric Sounding Network Based On Complete Complementary Code and Its Application”. Those comments are all valuable and very helpful for revising and improving our paper, as well as the important guiding significance to our research. We have studied comments carefully and have made correction which we hope meet with approval. The main corrections in the paper and the responds to the reviewer’s comments are as flowing:

1. Response to comment: The meaning of sequences and orders should be explained.

Response: As explained in the new manuscript, where M is the number of sequences in a sequence set, N is the number of subsequences of every sequence. As shown in Table 1, it is a sequence set with four sequences: A, B, C, D. And every sequence has four orders consisting of four subsequences like A1, A2, A3, A4 as an example.

2. Response to comment: Figure 2 is of a bad quality (low resolution).

Response: We are very sorry for our negligence. Figure 2 has been reproduced in a higher resolution. And the labels on axes have been enlarged to read comfortably.

3. Response to comment: It is not possible to read the letters A, B, C, D in Figure 4. They should be enlarged and at higher resolution. Also, what is the horizontal line in Figure 4. 

Response: We have made correction according to this comment. We are very sorry for the bad quality of some figures. In the new manuscript, Figure 4 has been reproduced in a higher resolution and the marks are clearer. And the horizontal line in Figure 4 where the point D located is the oblique echo of Es.

4. Response to comment: It should be (a) and (c), (b) and (e) [instead of (a) and (e), (b) and (c)].

Response: It is really true as suggested. We are very sorry for our incorrect writing. In the new manuscript, this mistake has been corrected.

5. Response to comment: I think that the performed inversion for oblique sounding was too simple to get reliable results in midpoints.  

Response: As described in the new manuscript, we use hybrid genetic algorithm (HGA) to carry out the inversion of the oblique sounding. It is an improved genetic algorithm (GA) based on simulated annealing algorithm (SA). It was firstly applied to the inversion of oblique ionogram by Song Huan et.al (Song, H.; Hu, Y.; Zhao. Z.; et al. Inversion of oblique ionograms based on hybrid genetic algorithm. Chinese J. Geophys. (in Chinese),2014,57(3); pp.703-714.). Because of the combination of the advantages of the two algorithms, HGA has high accuracy and efficiency (Wang, Z.; Cui, D. A hybrid algorithm based on genetic algorithm and simulated annealing for solving portfolio problem. International Conference on Business Intelligence and Financial Engineering, 2009.34; pp.106-109.). As proved by Song Huan, when it is used in oblique inversion, by comparing with the vertical sounding results at the reflection point, the parameters obtained by using this algorithm on the oblique inversion, for foF2 parameter as an example, show that HGA performs well in accuracy and stability.

And based on this, the regional maps (Figs. 9 & 10) in the old manuscript are constructed based on the foF2 parameters extracted from the inversion results by using Kriging (KG) algorithm which has been widely used to reconstruct maps of ionospheric parameters. When the spare date set is abundant, KG has good robustness. It can refer to: Jiang, C.; Zhou, C.; Liu, J.; et al. Comparison of the Kriging and neural network methods for modeling f oF2 maps over North China region. Advances in Space Research, 2015, 56(1); pp.38-46. And the description of this has also been added in the new manuscript.

6. Response to comment: As the Es layer was considered for vertical sounding (if I understand it well) and not for the oblique sounding, it is difficult to assimilate these together.

Response: It is really true as you suggested. Using different inversion models for vertical and oblique soundings may lead to a decrease in the accuracy of the results, which is also presented in the conclusion of Song Huan’s article. in the new manuscript, in consideration of this, we delete Figure 10 and its discussion. Meanwhile, referring to the article of Song, H. et al, the inversion results of foF2 parameters are quite accurate comparing with the vertical sounding results at the reflection point, although the Es layer existed. Therefore, we retain Figure 9, the assimilation maps of foF2 parameters, to illustrate the application prospects of this networking method. We believe that is also enough to show the superiority of this networking method. For the plasma frequencies, we plan to carry out more long-term and wide-ranging experiments after improving the inversion method in the next stage of work to obtain more meaningful results.

7. Response to comment: the profile calculated under assumption of propagation along the straight lines might differ from the real one. 

Response: In this article, we assume that the parameters of the ionosphere are similar in a small region. By comparing the two-way oblique ionograms of one propagation path, it confirms that the reflection points of the two-way propagation of the oblique soundings are very close. Therefore, we select three oblique ionograms from figure 7 to represent the oblique paths to calculate the profiles. And this has also been explained in the new manuscript. Of course, the obliquely propagating rays might be bent because of the refraction in the ionosphere. But for a large region, we believe that such a bend may not have much impact.   

Besides, Although the inversion of oblique ionogram is the ionospheric information at the midpoint of the propagation path, but referring to the comparison of the foF2 parameters obtained by the oblique inversion and the vertical sounding results at the reflection point in the article of Song, H. et al (in fact, she chose the geographic middle position of the stations), there is little difference between them. Therefore, we think this assumption will not cause significant errors in the results in this case.

8. Response to comment: To verify the inversion and to calculate the electron profile reliably from the oblique sounding, I’d like to suggest the authors a future experiment. If it is possible it would be useful to perform (at least for temporary time of several days) vertical sounding at a midpoint of oblique sounding. 

Response: We are very grateful for this proposal, and it has also been included in our next work plan. In fact, as described in the article of Song Huan et.al (Song, H.; Hu, Y.; Zhao, Z.; et al. Inversion of oblique ionograms based on hybrid genetic algorithm. Chinese J. Geophys. (in Chinese),2014,57(3); pp.703-714.), the comparison between the foF2 parameters obtained by using this algorithm on the oblique inversion and the vertical sounding results at the reflection point has been made. And the results verify the accuracy of this method. Next, we plan to conduct similar experiments to further improve the inversion accuracy, to obtain other meaningful ionospheric parameters, and to verify the ability of this networking method to observe TIDs.

9. Response to comment: The reference [9] is missing in the text.

Response: We are very sorry for this mistake. It has been corrected in the new manuscript.

10. Response to comment: Lines 38-39, “…ionospheric oblique sounding is an effective substitute resolution”. I do notunderstand this, please reformulate.

Response: It has been reformulated as”In many cases, ionospheric oblique sounding is an effective substitute solution”.

11. Response to comment: Line 54, it is unusual that the reference to Figure 4 appears before the reference to Figure 1. I suggest removing the reference to Figure 4 here and/or reformulating.

Response: We reformulate this sentence as “However, when the distance between two stations is not suitable, the low-frequency part of the oblique echoes can easily overlap with the vertical echoes or its multi-hop, which will be well illustrated below of the present study.”

12. Response to comment: Figure 1 caption, „0kHz ...0us“, remove the units or insert space between 0 and the unit. (likely us means )

Response: We insert space between 0 and the unit in the new manuscript.

Response to comment: Line 161, „all“, move the word „all“ to the beginning of the sentence or remove it.

Response: It has been reformulated as “All the antennas for transmitting and receiving have the omnidirectional patterns. The specific sounding parameters are shown in Table 2.”.

13. Response to comment: Line 163, „In our experiments, we carried out two cases“ I suggest replacing by „We carried outtwo experiments“ .

Response: It has been reformulated as ” There are two experiments have been carried out.”

14. Response to comment: Table 2, us replace by . Also, specify if the pulse width correspond to 1 bit or to something else.

Response:  It has been reformulated as below.

Pulse width

25.6(/bit)

15. Response to comment: Line 224, „In the second case, as a clear contrast can be made...“ I suggest reformulating, e.g., by„In the second case, the interference is not a problem as demonstrated in Figure 5 which shows...“

Response: It has been reformulated as” In the second case, the interference is not a problem as demonstrated in Figure 5 which shows the vertical ionograms recorded at 22:06 BJT on 31 August 2018 in which (a) is the vertical ionogram of Leshan station, (b) is the result of Ningqiang, and (c) is for Wuhan”.

16. Response to comment: Line 246, „inongrams“ replace by „ionograms“. Also delete „the effect of “ in this line.

Response: We are very sorry for our negligence. And the incorrect writing has been corrected.

17. Response to comment: Line 330 and 331, 0.14 MHz and 0.05 MHz is likely wrong here. I think that the plasma frequency is much higher at these altitudes.

Response: We are sorry for that we didn't make it clear here. In the new manuscript, it has been reformulated as “At the height of 240km, the plasma frequency near Wuhan is about 0.14MHz higher than the surrounding area. At the height of 250km, it is about only 0.05MHz higher than the surrounding area.”

18. Response to comment: Line 352, „enhancing“what it means here, in which sense? Reformulate.

19. Response: As described in the new manuscript, we believe this network method can enhance the global navigation satellite systems (GNSS) by providing regional ionospheric reference information.

Besides, some other revises are also highlighted in the new manuscript.

Sincerely, we feel very grateful to the editors and reviewers for the kind work and pertinent comments. These comments provide us with a good guidance both about the scientific research and writing, which is of great benefit to our improvement. On behalf of my co-authors, we would like to express our great appreciation to the editors and reviewers.

Sincerely,

Tongxin Liu

Reviewer 2 Report

I think the work presented by the authors of this paper is very interesting, and may prove an important development in ionosonde technology. In general the manuscript is of high quality, and almost ready for publication.

However, there is one point that should certainly be addressed: it should be made clear how the ionogram inversion is done, and how the regional maps (Figs. 9 & 10) are constructed. I understand this is not the main point of this paper, so the description could be brief if good references are provided. This is mostly important because the the maps show some features that look like artifacts (especially clear in Fig. 10f).

Besides the above, I have a few smaller comments:

In lines 6-56, a brief comment is made about the operations of the Lowell DPS4D ionosondes, saying that there is a difficulty in distinguishing vertical and oblique echoes since they may appear overlapping each other in the ionogram. However, it should be noted that the DPS4D uses an array of four receiving antennas instead of one, precisely to allow detecting the direction from which an echo is being received. Thus, for the DPS4D the problem of distinguishing between vertical and oblique echoes does not exist, albeit at the cost of requiring four receiving antennas instead of one (see chapter 2 of the DPS4D manual, freely available here: http://digisonde.com/dps-4dmanual.html). This should be  clarified in the text. The prime advantage of the here described system seems to be that only a single antenna is required, while the main disadvantage, as identified by the authors, is the order of the used CCCs.

Some figures are of rather low quality, and difficult to read. Figure 2 should be reproduced in a higher resolution, and the marking on Figure 4 be made more legible (for instance by using an different color).

In Figures 6 and 8, both the vertical and horizontal axes are inconsistent between panels. It is easier for the reader to interpret these figures if all panels use the same scales.

Author Response

Dear Editors and Reviewers:

Thank you for your letter and for the reviewers’ comments concerning our manuscript entitled “A Novel Ionospheric Sounding Network Based On Complete Complementary Code and Its Application”. Those comments are all valuable and very helpful for revising and improving our paper, as well as the important guiding significance to our research. We have studied comments carefully and have made correction which we hope meet with approval. The main corrections in the paper and the responds to the reviewer’s comments are as flowing:

1. Response to comment: How the ionogram inversion is done?

Response: The vertical ionograms are inversed based on QPS model and the oblique ionograms are inversed using hybrid genetic algorithm (HGA) based on QP model and the monolayer pattern. The specific inversion process and the comparison between the oblique inversion and the inversion of the vertical sounding results at the reflection point can refer to: Song, H.; Hu, Y.; Zhao. Z.; et al. Inversion of oblique ionograms based on hybrid genetic algorithm. Chinese J. Geophys. (in Chinese),2014,57(3); pp.703-714. The results of Song Huan ‘s experiments show that performs well in accuracy and stability.

2. Response to comment: How the regional maps (Figs. 9 & 10) are constructed?

Response: The regional maps (Figs. 9 & 10) are constructed based on the foF2 parameters and the plasma frequency parameters extracted from the inversion results by using Kriging (KG) algorithm which has been widely used to reconstruct maps of ionospheric parameters. When the spare date set is abundant, KG has good robustness. It can refer to: Jiang, C.; Zhou, C.; Liu, J.; et al. Comparison of the Kriging and neural network methods for modeling f oF2 maps over North China region. Advances in Space Research, 2015, 56(1); pp.38-46. But in the new manuscript, in consideration of the difference of the models used for the vertical and oblique inversions, it may be inappropriate to assimilate the plasma frequencies together. Therefore, we delete Figure 10 and its discussion. Meanwhile, referring to the article of Song, H. et al, the inversion results for foF2 parameters are quite accurate comparing with the vertical sounding results at the reflection point. Therefore, we retain Figure 9 to illustrate the application prospects of this networking method. We believe that is also enough to show the superiority of this networking method. For the plasma frequencies, we plan to carry out more long-term and wide-ranging experiments after improving the inversion method in the next stage of work to obtain more meaningful results.

3. Response to comment: It should be clarified in the text that for the DPS4D the problem of distinguishing between vertical and oblique echoes does not exist.

Response: It is really true as the reviewer suggested that we should clarify that the DPS4D can distinguish the vertical and oblique echoes by detecting the direction from which an echo is being received with an array of four receiving antennas. We are very grateful for your kind reminder. It has been corrected and highlighted in the second paragraph of the Introduction. For the method described in the manuscript, as highlighted in the third paragraph of the Introduction, relaying on the orthogonality between the CCC sequences, every ionosonde can also receive the oblique signals transmitted by the others without the need of the antenna array. It is one of the advantages of this network method.

4. Response to comment: Some figures are of rather low quality.

Response: We are very sorry for our negligence. According to this comment, figure 1,2,4,5,6,7,8 have been reproduced in a higher resolution. And the labels on axes have been enlarged to read comfortably.

5. Response to comment: In Figures 6 and 8, both the vertical and horizontal axes are inconsistent between panels.

Response:  Both the vertical and horizontal axes are unified. In Figures 6, the frequency range is limited at 0-8MHz while the altitude is limited 60 to 300 km. In Figures 8, the frequency range is limited at 0-8MHz while the altitude is limited 200 to 300km. And these are highlighted in the new manuscript.  

Besides, some other revises are also highlighted in the new manuscript.

Sincerely, we feel very grateful to the editors and reviewers for the kind work and pertinent comments. These comments provide us with a good guidance both about the scientific research and writing, which is of great benefit to our improvement. On behalf of my co-authors, we would like to express our great appreciation to the editors and reviewers.

Sincerely,

Tongxin Liu

Round 2

Reviewer 1 Report

see the attached file please.

Author Response

Dear Editors and Reviewers:

Thank you for your letter and the further comments concerning our manuscript entitled “A Novel Ionospheric Sounding Network Based On Complete Complementary Code and Its Application”. We have studied comments carefully and the specific corrections and the responds are as follows:

1. Response to comment: Comparison of Figure 6 and 8. It should be discussed why in Figure 6 (vertical sounding) is nonzero electron density (frequency) between approximately 90 and 220 km, whereas the electron density (frequency) is zero in this range in Figure 8 (oblique). Is it because the Es and E layer was neglected in oblique sounding? 

Response: In Figure 8, the Es and E layer are not involved in inversion. With a view that the oblique echoes of the E layer and the Es layer are not easy to distinguish in the ionograms. It is difficult to obtain the correct profiles of E or Es layer. Therefore, in this paper, we set the F layer parts as the main objects of the oblique inversions. Oblique inversions are also conducted only for F layer.

2. Response to comment: The authors state that the oblique inversion works good for foF2
even if different models were used (no E layer for oblique inversion considered). It would be
good to specify (estimate) the reliability (accuracy) in numbers. The statement “quite reliable”
is vague.

 Response: We are very grateful for this proposal. It is really true as you suggested, that the previous statements are indeed somewhat vague. In view of this, we have added an explanation in the manuscript. As described in the new manuscript, in Song Huans experiments, referring to the vertical sounding results of the reflection point, 84.62% of the oblique inversion results are within the error of 0.4MHz.

3. Response to comment: Line 173, There are two experiments…-> Two experiments…

Response: Thank you very much for your suggestion in writing. We have corrected it.

All new revisions are highlighted in green in the new manuscript.

Once again, Id like to express our great appreciation to you, for your patient and meticulous work, which is very helpful to our article. Thank you very much.

Sincerely,

Tongxin Liu
